# An unbiased comparison of 14 epigenetic clocks in relation to 174 incident disease outcomes

Christos Mavrommatis[1,2], Daniel W. Belsky [3], Kejun Ying[4], Mahdi Moqri [4], Archie Campbell [1,5], Anne Richmond[1], Vadim N. Gladyshev [4], Tamir Chandra [2,6,7], Daniel L. McCartney[1,8,9] & Riccardo E. Marioni [1,9] ✉

Epigenetic Clocks have been trained to predict chronological age, healthspan and lifespan. Such clocks are often analysed in relation to disease outcomes – typically using small datasets and a limited number of clocks. Here, we present a large-scale ($n = 18,859$), unbiased comparison of 14 widely used clocks as predictors of 174 incident disease outcomes and all-cause mortality over 10-years of follow up. Second- and third-generation clocks significantly out-perform first-generation clocks, which have limited applications in disease settings. Of the 176 Bonferroni significant ($P < 0.05/174$) associations from fully-adjusted Cox regression models controlling for lifestyle and socio-economic measures, there are 27 diseases (including primary lung cancer and diabetes) where the hazard ratio for the clock exceeds the clock's association with all-cause mortality. Furthermore, for 32 of the 176 findings, adding the clock to a null classification model with traditional risk factors significantly increases the classification accuracy by >1%. However, there is minimal evidence for interactions between the clocks and sex or smoking (ever/never) status. Second- and third-generation epigenetic clocks show promise for disease risk prediction, particularly in relation to respiratory and liver-based conditions.

We all age at the same chronological rate; everyone gets a year older every 12 months. But we experience the passage of time differently. Some of us maintain our health into our later years, while others develop chronic disease and disability by midlife. Defining the biological differences that underlie this heterogeneity is complex. There is no gold standard measurement of biological aging[1]. However, there are now a number of candidate biomarkers of aging derived from integrating high-dimensional molecular data with machine-learning methods that have accumulated substantial validation evidence[2]. Among these, the best studied are a family of DNA methylation (DNAm) algorithms known as epigenetic clocks[3].

The first generation of epigenetic clocks were developed by comparing DNA methylation data between older and younger people to develop algorithms predictive of chronological age e.g., the Horvath and Hannum clocks[4,5]. A second generation of epigenetic clocks were subsequently developed by modelling differences between individuals

[1]Institute of Genetics and Cancer, University of Edinburgh, Edinburgh, UK. [2]Robert and Arlene Kogod Center on Aging, Mayo Clinic, Rochester, MN, USA. [3]Department of Epidemiology, Butler Columbia Aging Center, Mailman School of Public Health, Columbia University, New York, NY, USA. [4]Department of Medicine, Brigham and Women's Hospital, Harvard Medical School, Boston, MA, USA. [5]Centre for Medical Informatics, Usher Institute, University of Edinburgh, Edinburgh, UK. [6]Department of Biochemistry and Molecular Biology, Mayo Clinic, Rochester, MN, USA. [7]Department of Quantitative Health Sciences, Mayo Clinic, Rochester, MN, USA. [8]bioXcelerate, Optima Partners, Edinburgh, UK. [9]These authors contributed equally: Daniel L. McCartney, Riccardo E. Marioni. ✉e-mail: riccardo.marioni@ed.ac.uk

in mortality risk and phenotypes that track general health[6,7]. Finally, a third generation was developed by modelling differences between individuals of the same chronological age in their rate of multi-organ system deterioration[8]. The later generation clocks are therefore trained to predict complex outcomes, compared to chronological age. They often include two-step processes whereby DNAm proxies of health-related biomarkers (e.g., smoking, protein levels) are considered in place of individual DNAm CpG sites that are typically used to derive first-generation clocks. While this incorporation of broader biological information may enhance the utility for clinical outcome prediction, it also complicates interpretability compared to first-generation clocks, which may remain more suitable for investigating the mechanisms underlying cellular aging.

A key question in evaluating the clinical utility of epigenetic clock biomarkers of aging is their capacity to predict the future incidence of aging-related disease. However, while there are now a large number of studies reporting clock associations with different disease outcomes in a range of datasets[9,10], no large-scale systematic comparison of many clocks predicting many diseases has been conducted.

Here, we provide an unbiased assessment of 14 clocks, including the most widely studied first-, second- and third-generation clocks (Hannum, Horvath, PhenoAge, GrimAge and DunedinPACE, respectively; Table 1) in relation to 10-year onset of a comprehensive set of 174 disease outcomes within one of the world's single-largest DNAm datasets, the Generation Scotland cohort ($n = 18,859$)[11]. There was a minimum of 30 incident cases per disease (median = 108, max = 1527 cases for hypertension, see Supplementary Data 1).

## Results

### Cox regression of 14 epigenetic clocks and 174 diseases
Cox proportional hazards regression was run for each clock-disease pairing, first adjusting for just age and sex (Supplementary Data 2) followed by a fully-adjusted model that also controlled for body mass index, smoking, alcohol consumption, education, and socioeconomic deprivation (Supplementary Data 3). Interaction models were considered for both the basic and fully-adjusted models to test for differences by smoking status (ever/never smoked) or sex (Supplementary Data 2). Age, estimated cell proportions and relatedness were regressed out of each clock prior to the Cox regression analyses. The fully-adjusted covariate setup might directly capture

information that is included in the clocks, e.g., a DNAm proxy for smoking is included in the construction of GrimAge, a predictor of mortality[7]. However, these covariates are easy to assess and commonly collected in clinical settings. Thus, in terms of disease prediction, it is valuable to note what added information epigenetic clocks might contribute. A total of 176 Bonferroni significant associations were found for 13 of the clocks across 57 diseases in the fully-adjusted models ($P < 2.9 \times 10^{-4}$). In addition to the Cox regressions, logistic regression 10-year classification was conducted. Differences in AUC were tested for between nested models (first adjusting for the full set of aforementioned covariates and then adding in an epigenetic clock).

There were between 12 (Lin clock) and 72 (both GrimAge v2 and DunedinPACE) statistically significant associations ($P < 0.05/174$) in the age- and sex-adjusted Cox regression models (Table 1 and Supplementary Data 2).

An R Shiny app displays the results from both the fully-adjusted Cox and logistic regression analyses (https://shiny.igc.ed.ac.uk/Epigenetic_Clock_and_Disease_Association/). Users can visualise findings for specific or multiple clocks and disease outcomes or diseases within a grouping, e.g, psychiatric or cardiovascular traits.

In the fully adjusted Cox regression models, there were 9 Bonferroni significant disease associations for the first-generation clocks, representing ~5% of all significant findings. Across all 174 diseases that were considered, and using GrimAge v1 as the reference category, the average log hazards (Fig. 1, Supplementary Data 4, Supplementary Data 5) of the first-generation clocks were around 50% smaller in magnitude ($P \leq 6.8 \times 10^{-12}$). Smaller effect sizes were also noted for PhenoAge ($P = 2.9 \times 10^{-3}$) and DNAm telomere length ($P < 6.2 \times 10^{-6}$). Of the second-generation clocks, smaller effect sizes were observed for Zhang10, larger effect sizes were seen for DunedinPACE and GrimAge v2 while there was no significant difference with DunedinPoAm.

### Cox regression of 14 epigenetic clocks and mortality
To help contextualise the magnitude and significance of the clock-disease associations, we first ran survival models for 10-year all-cause mortality ($n_{cases} = 842$; Supplementary Data 6)—one of the best-established health outcomes to be related to epigenetic age acceleration[12]. The largest and most significant association was with GrimAge v2 (Hazard Ratio (HR) per SD of age acceleration = 1.54, 95% CI [1.46, 1.62], $P = 7.1 \times 10^{-62}$). All clocks had significant associations at

**Table 1 | Overview of the 14 epigenetic clocks considered as disease risk predictors**

| Clock | Generation | Outcome | Tissue | N CpGs in Clock | N disease associations (Cox models) | | | |
|---|---|---|---|---|---|---|---|---|
| | | | | | Age and sex adjusted | | Fully adjusted | |
| | | | | | P < 0.05 | P < 0.05/174 | P < 0.05 | P < 0.05/174 |
| Hannum (2013)[4] | First | Age (years) | Blood | 71 CpGs | 36 | 29 | 4 | 1 |
| Horvath (2013)[5] | First | Age (years) | Multi- tissue | 353 CpGs | 27 | 17 | 2 | 1 |
| Lin (2016)[27] | First | Age (years) | Blood | 100 CpGs | 17 | 12 | 2 | 1 |
| Zhang10 (2017)[28] | Second | Mortality | Blood | 514 CpGs | 84 | 60 | 38 | 20 |
| PhenoAge (2018)[6] | Second | Phenotypic Age | Blood | 513 CpGs | 76 | 54 | 31 | 22 |
| Horvath Skin & Blood (2018)[29] | First | Age (years) | Skin and Blood | 391 CpGs | 40 | 28 | 6 | 3 |
| GrimAge v1(2019)[7] | Second | Mortality | Blood | 1030 CpGs | 102 | 68 | 57 | 29 |
| DNAmTL (2019)[30] | NA | Telomere Length | Blood, Adipose | 146 CpGs | 60 | 47 | 22 | 5 |
| Dunedin PoAm38 (2020)[31] | Third | Pace of Aging | Blood | 46 CpGs | 95 | 58 | 47 | 21 |
| Dunedin PACE (2022)[8] | Third | Pace of Aging | Blood | 173 CpGs | 103 | 72 | 60 | 33 |
| GrimAge v2 (2022)[32] | Second | Mortality | Blood | 1030 CpGs | 105 | 72 | 67 | 37 |
| YingCausAge (2024)[33] | First | Age (years) | Blood | 586 Causal CpGs | 27 | 19 | 1 | 0 |
| YingDamAge (2024)[33] | First | Age (years) | Blood | 1090 Damaging CpGs | 28 | 19 | 6 | 2 |
| YingAdaptAge (2024)[33] | First | Age (years) | Blood | 1000 Protective CpGs | 28 | 20 | 2 | 1 |

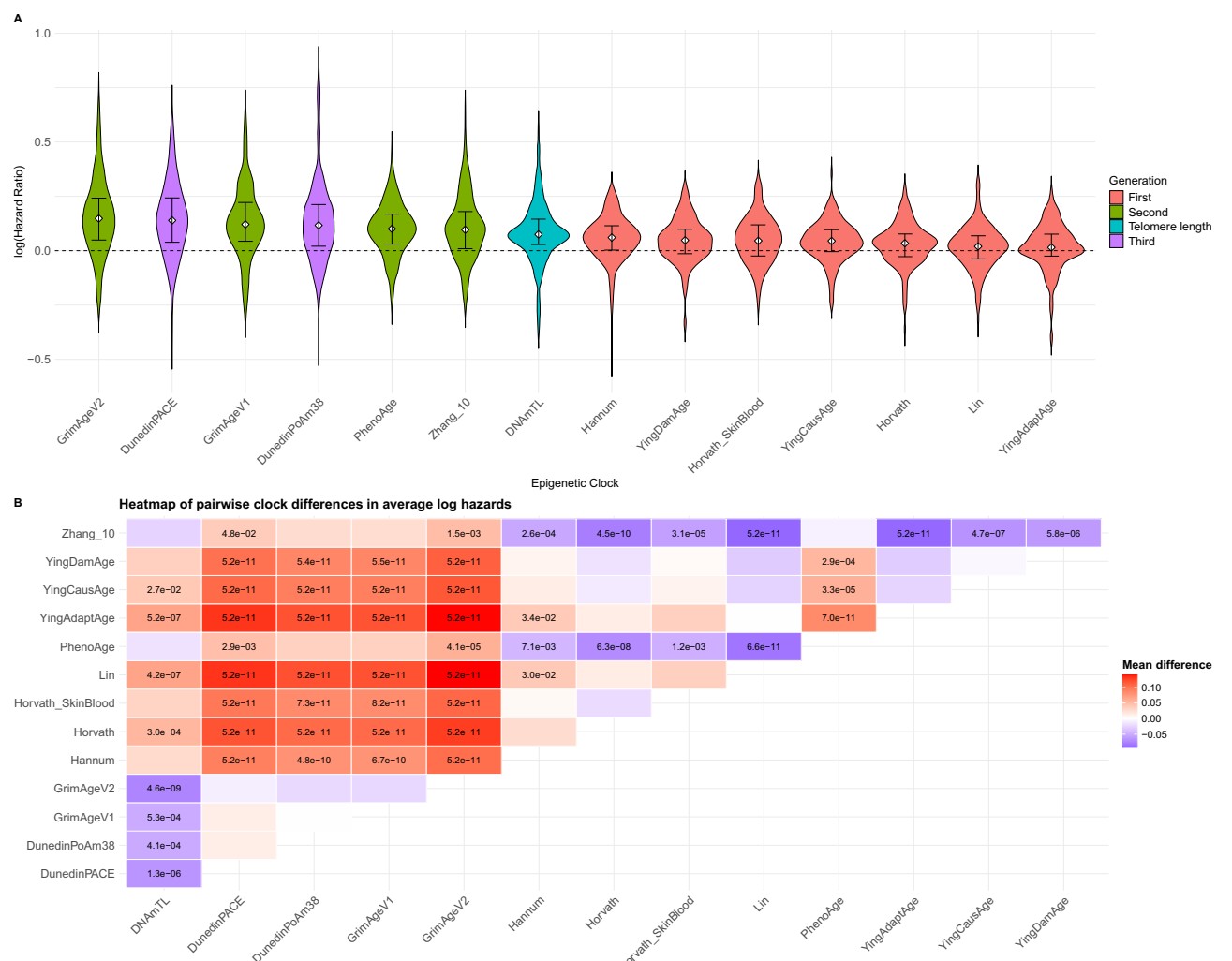

**Fig. 1 | Comparison of epigenetic clock effect sizes across 174 incident disease outcomes. A** Distribution of log hazard ratios for each epigenetic clock across all 174 incident disease outcomes. Clocks are presented in descending order of the median effect size. Median and interquartile range presented within the violins. First-generation clocks are highlighted in pink, second-generation in green, third-generation clocks in purple and telomere length (where log HR effect sizes have been multiplied by −1 for visual display purposes) in turquoise. **B** Comparison of average log hazard ratios across 174 disease outcomes across pairs of epigenetic clocks (two-sided tests). Telomere length log hazards have been multiplied by −1 prior to comparisons. The mean differences are presented as x axis clock minus y axis clock e.g., there is a negative difference between DNAmTL and DunedinPACE, indicating the mean log hazards for the latter are larger. P values are provided for nominally significant differences (P < 0.05).

$P < 0.05$ apart from Horvath's skin and blood clock, Ying's AdaptAge and Lin's age predictor. Using logistic regression, an AUC of 0.851 was obtained for the model with covariates only. This increased by up to 0.014 (1.4%) upon the addition of GrimAge v2 (AUC = 0.865) as a covariate.

There were 27 unique disease outcomes where the clock-disease hazard ratio was Bonferroni-significant and exceeded the magnitude of the corresponding clock-mortality association (Supplementary Data 7). A diverse set of diseases were highlighted, however there was a strong focus on respiratory/smoking-related and liver-related outcomes, including primary lung cancer (HR$_{GrimAgev1}$ per SD = 1.56 [1.42, 1.72], $P = 5.3 \times 10^{-19}$) and cirrhosis (HR$_{GrimAgev2}$ = 1.86 [1.57, 2.21], $P = 8.9 \times 10^{-13}$). Also of note, were associations with diabetes (HR$_{DunedinPACE}$ = 1.44 [1.33, 1.57], $P = 9.6 \times 10^{-19}$), Crohn's disease (HR$_{PhenoAge}$ = 1.39 [1.19, 1.64], $P = 4.7 \times 10^{-5}$) and delirium (HR$_{Zhang10}$ = 1.44 [1.23, 1.68], $P = 6.7 \times 10^{-6}$).

### Age and sex stratified clock-disease analyses
In the fully adjusted sex- and smoking-stratified analyses and interaction models, there were only two significant findings from analyses

where the number of cases per stratum exceeded 30 (Supplementary Data 2): the associations between GrimAge v1 and v2 and the risk of post-viral fatigue syndrome (HR per SD of GrimAge v1 in ever smokers = 1.51 compared to 0.81 in never smokers, P$_{interaction}$ = 6.2 × 10$^{-5}$).

### Logistic regression for 10-year classification
Logistic regression models were then run for all clock-disease analysis using a null model (covariates only) and a full model where a clock was added as a predictor of interest. After filtering to the 176 clock-disease associations that were Bonferroni significant in the Cox models, there were 32 instances where the AUC improvement between the null and full model was greater than 0.01 and nominally significant at $P < 0.05$ (Fig. 2 and Supplementary Data 8).

Again, several clocks were linked to respiratory/smoking-related disease outcomes, including COPD, lung cancer and respiratory failure.

### Discussion
Our findings clearly indicate that second- and third-generation epigenetic clocks should be prioritised for disease association studies.

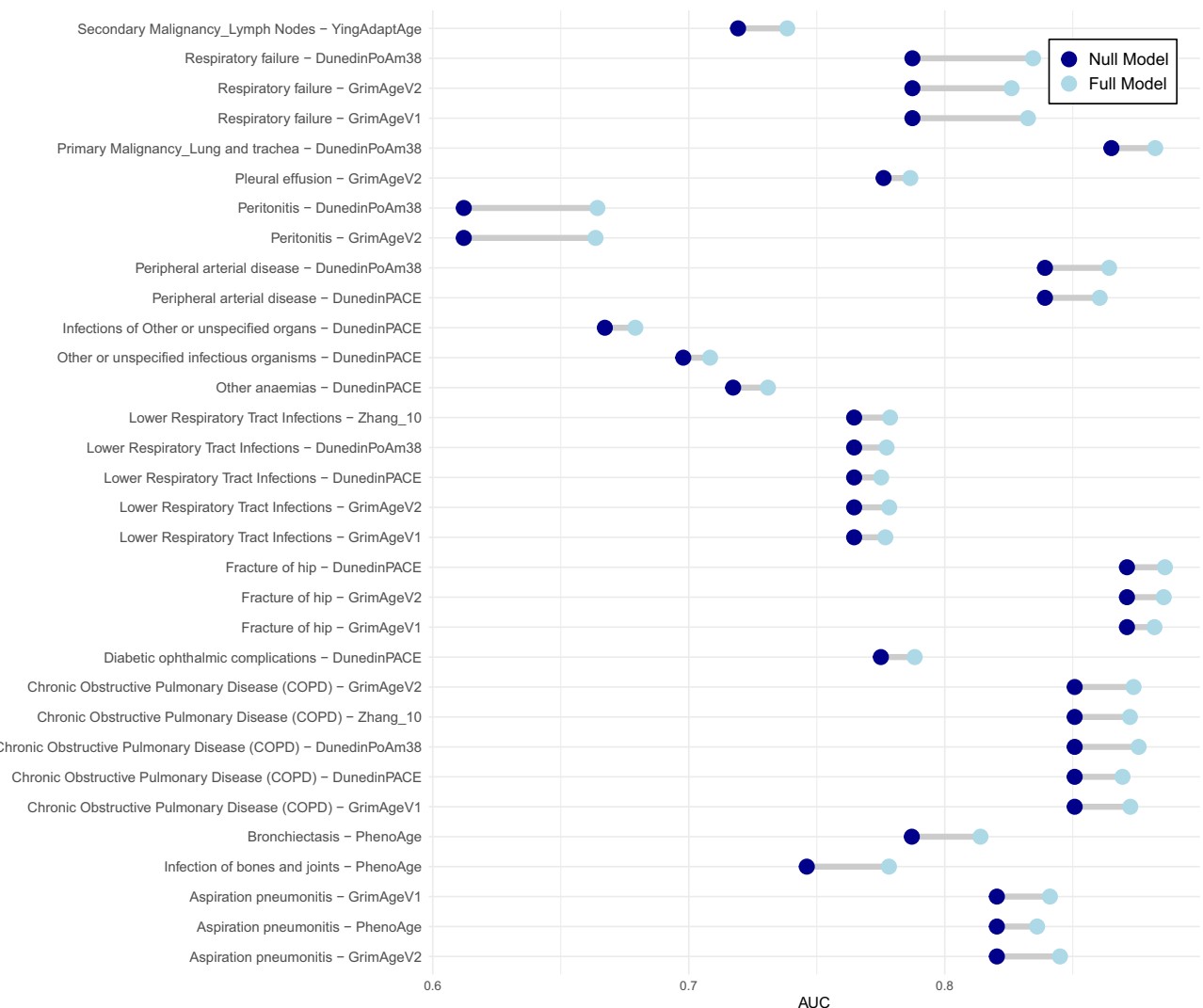

**Fig. 2 | Significant clock-disease associations with AUC improvements > 1%.** Area Under the Curve (AUC) increments for disease–clock associations where the hazard ratio from the Cox regression was Bonferroni significant ($P < 0.05/174$), and the improvement to the AUC upon adding the clock to the null model was > 0.01 and nominally significant ($P < 0.05$) in a comparison of the nested models (two-sided tests). Null model covariates include: age, sex, education, alcohol, smoking, BMI and deprivation. The full models also included the relevant epigenetic clock.

These clocks showed particularly strong links to respiratory and liver-related disease outcomes, including primary lung cancer and cirrhosis. Furthermore, the findings are present after adjusting for key risk factors such as deprivation, self-reported smoking behaviour and alcohol consumption as covariates. There was minimal evidence for differences in the clock-disease associations by sex or smoking status.

The 162 Bonferroni significant second- and third-generation clock disease associations were evenly spread across the clocks, with a maximum of 37 associations from GrimAge v2 i.e., there is no clear "best" clock for pan-disease analyses. The construction of both v1 and v2 of GrimAge includes a DNAm surrogate for smoking, which may help to drive downstream associations with respiratory diseases. Similarly, the inclusion of longitudinal changes in BMI, waist-hip ratio and HbA1c in DunedinPACE may contribute to its predictive utility for diabetes.

Our study contains certain limitations. For example, we related DNAm profiling from whole-blood samples against multi-tissue diseases and non-blood tissue specific diseases in a Scottish-based cohort. We also restricted our analyses to 14 epigenetic clocks that are presented on the Biolearn platform. Our data analysis did not exhaustively adjust for covariates that might track disease risk and multi-morbidity, such as family history of disease, genetic risk scores (polygenic scores), medication usage, social interactions and blood pressure. However, bespoke covariate setups are not feasible when studying such a broad range of disease outcomes. Smoking status and alcohol consumption over the past week were self-reported and may therefore carry biases. Future studies could consider the addition of epigenetic proxies for these measures as covariates. It is also crucial to highlight that epigenetic clocks do not provide insights into the mechanisms of action of diseases. Despite the longitudinal data utilised in this study to compute incident disease outcomes, serial bio-samples were not available. Replication of our findings using longitudinal samples across diverse cohorts is also needed.

In our effort to summarise associations across a very large number of individual diseases, we were limited in our ability to explore heterogeneity. While we consider differences by smoking status and sex, we have not explored variation across different periods of follow-up, strata of co-morbidities, or stages of disease. These represent important future directions for follow-up studies, as do investigations of clock associations with metrics of aggregate disease burden, such as indices of comorbidity. Future research should also consider the training and testing of disease-specific clocks. For example, we have

previously shown that a DNAm predictor for incident Type 2 Diabetes augments models containing traditional risk factors[13]. Finally, our disease phenotyping approach, whilst consensus-based, might be too broad in some instances, e.g., the all-cause dementia phenotype captures both Alzheimer's disease and vascular dementia, as well as other, less common dementia sub-types.

Here, we performed a comprehensive and unbiased comparison of 14 epigenetic clocks as putative biomarkers for 174 incident disease outcomes. We identified 176 disease associations across 13 clocks with 57 unique diseases. By anchoring our findings to well-established associations with all-cause mortality, we highlighted multiple associations with large effect sizes. We also focus on outcomes where classification gains may lead to clinical impact. These results form a starting point for the targeted selection of epigenetic clocks for consideration in clinical risk prediction models.

## Methods

### Ethics and consent
All components of Generation Scotland received ethical approval from the NHS Tayside Committee on Medical Research Ethics (REC Reference Number: 05/S1401/89). All participants provided broad and enduring written informed consent for biomedical research. Generation Scotland has also been granted Research Tissue Bank status by the East of Scotland Research Ethics Service (REC Reference Number: 15/0040/ES), providing generic ethical approval for a wide range of uses within medical research. This study was performed in accordance with the Helsinki declaration.

Self-reported biological sex was considered in this study–this was cross-referenced via genetic data. The distribution of sex by disease outcome is provided in Supplementary Data 1. Analyses were conducted on the full cohort to maximise statistical power; an interaction with sex and sex-stratified models were also considered. Participants did not receive compensation for participation.

### Generation Scotland Cohort
Generation Scotland (GS) is a family-based research study comprised of 24,084 volunteers[14]. Volunteers joined the cohort between 2006 and 2011 with the majority of recruitment taking place via invitation from the individuals' general practitioners (GPs). Initially, individuals aged 35–65 years were recruited from across five geographic areas (Glasgow, Aberdeen, Dundee, Ayr and Arran). They were then encouraged to recruit family members living in Scotland to join the study. This resulted in a baseline cohort aged between 17 and 99 years. All individuals were invited to complete questionnaires and to attend a baseline clinic visit for further questionnaires and testing, resulting in a detailed collection of lifestyle, clinical, health, cognitive and socio-demographic data. The majority also agreed to provide biosamples, including blood, from which DNA has been extracted and profiled for genetic and epigenetic data. Most individuals also provided consent for researchers to access their medical records via data linkage to primary and secondary care codes.

### DNA methylation profiling
Methylation data have been quantified for 18,869 GS volunteers at a total of 851,610 genomic (CpG) sites after quality control via the Illumina EPIC850k array[11]. Following 10 study withdrawals, the analysis sample size was 18,859. Epigenetic age and age acceleration estimates were derived via the Biolearn platform. Biolearn is an open-source python library that enables easy and versatile analyses of biomarkers of aging data. It provides tools to easily load data from publicly available sources and contains reference implementations for common aging clocks such as the Horvath clock, DunedinPACE, and many others that can easily be run in only a few lines of code[15].

Age acceleration for each clock/predictor was calculated as the residual from a linear mixed model where the clock/predictor was regressed on age and estimated cell proportions derived from the Biolearn platform using the Salas et al. approach[16]. To avoid collinearity (as the cell proportions sum to one), basophil proportion was not included as a covariate. A pedigree kinship matrix was included as a random effect to control for the family structure present in the cohort. The regression models were run using the lmekin function from the coxme R package (2.2-22)[17]. As there are multiple approaches to estimating white cell proportions from DNA methylation data, a sensitivity analysis was conducted where each clock was adjusted for proportions derived from the EpiDISH R package[18] (version 2.22.0; reference panel set to "cent12CT.m", method = "RPC"; basophil proportion was excluded to avoid collinearity as the sum of the proportions is one). The correlation between these residuals and those used in the analyses ranged between 0.90 (Zhang clock) to 0.99 (Ying-DamAge), suggesting minor differences between the two approaches.

### Disease coding
Using a list of consensus definitions based on primary and secondary care codes[19] 308 disease outcomes were derived from the electronic health records data in Generation Scotland from January 1980 to April 2022. Despite having volunteer consent to access primary care data, these were only accessible for ~40% of the cohort due to consent constraints with individual GP surgeries (the data holders). These data were subsequently filtered to only consider the first diagnosis for each disease, which could be made in either a primary or secondary care setting. Those with a diagnosis prior to joining the study were excluded from downstream analyses for that specific disease outcome, in order to ensure investigation of incident as opposed to prevalent disease outcomes. Not all diseases were observed in the cohort. The latest date of linkage with primary and secondary healthy records was April 2022. Filtering to diseases with at least 30 cases over the first 10-years of follow-up resulted in 174 outcomes for the main analyses. A list of the 174 diseases observed and the number of incident events (events occurring after the blood draw) along with mean age at diagnosis are presented in Supplementary Data 1.

### Mortality Records
Mortality records were obtained via Community Health Index linkage to the National Records of Scotland. These data formed part of our censoring criteria where individuals either survived and remained unaffected by a disease or died during the follow up period to April 2022.

### Covariates
Demographic, lifestyle and health variables were considered as covariates in the regression models with each clock and disease outcome. These included: age, sex, body mass index (BMI, calculated as weight in kg divided by squared height in metres), smoking pack years (calculated by self-reported packs per day multiplied by number of years as a smoker), alcohol intake, which was assessed as units consumed over the last week in a self-report questionnaire, years of education (an ordinal variable with 11 self-report categories: (1) 0 years, (2) 1–4 years, (3) 5–9 years, (4) 10–11 years, (5) 12–13 years, (6) 14–15 years, (7) 16–17 years, (8) 18–19 years, (9) 20–21 years, (10) 22–23 years, (11) 24 or more years), and an area-based rank of socioeconomic deprivation (Scottish Index of Multiple Deprivation, SIMD). SIMD measures relative deprivation across 6976 data zones, calculated across seven domains: income, employment, education, health, access to services, crime and housing.

K-nearest neighbours' imputation–using k = 10 and a maximum missingness per person of 4/7 of variables– was used to generate complete data for covariates via the impute package in R (version 1.80.0)[20]. There was complete data on age and sex with a range of missingness for the other variables (SIMD: $n = 1152$, BMI: $n = 120$, Education: $n = 1012$, Pack years: $n = 395$, Alcohol: $n = 1734$). Natural log

transformations were applied to alcohol and pack years (adding a constant of 1 to all values to account for non-drinkers/smokers) and BMI. All covariates were then standardised to mean zero and unit variance prior to imputation.

## Association between age acceleration and disease outcomes

For the primary analysis, Cox proportional hazards regression models were run using the survival package in R (version 3.8-3)[21], adjusting for the aforementioned covariates, and were used to relate age acceleration for each epigenetic clock to 10-year onset for the list of 174 incident disease outcomes. The analyses were subset to the most frequently affected sex where the proportion of male or female cases was >90% for any given disease. A total of 174 models were run for each of the 14 clocks, giving 2436 models in total. $P$ value filtering was conducted per individual clock-regression model, resulting in a Bonferroni threshold of $P < 0.05/174 = 2.9 \times 10^{-4}$. A further set of analyses were run where time to all-cause mortality was the outcome.

Additional analyses included basic-adjusted Cox models (controlling only for age and sex) as well as sex- and smoking (ever/never)-stratification and interaction testing for both the primary analysis (fully-adjusted Cox models) and the basic-adjusted Cox models. When stratifying by smoking or testing the interaction formally, the continuous pack years variable was excluded as a covariate.

The proportional hazards assumption was tested by examining the Schoenfeld residuals. A test of the residuals for both the overall model and the predictor variable of interest (age acceleration) was performed using the cox.zph function in the survival package. This was done for both the basic- and fully-adjusted Cox models as well as for the stratified analyses (male, female, ever smoker, never smoker).

A formal comparison of the log Hazard Ratios for the clocks across all 174 disease outcomes in the fully adjusted model (primary analyses) was then conducted using a linear mixed effects model (using the lme4 and lmerTest R packages, versions 1.1-35.4 and 3.1-3, respectively)[22,23]. The logHR was the outcome while the clock was the predictor (GrimAge v1 was the reference level) and disease was included as a random effect on the intercept. The emm function from the emmeans R package (version 1.10.2)[24] was then used to obtain the marginal means for each clock, prior to formal pairwise comparisons with a Tukey adjustment to account for multiple comparisons.

In addition to the Cox models, logistic regression was used to generate area under the curve (AUC) estimates for 10-year disease classification. Null models contained the aforementioned covariates, while a full model also included an age acceleration measure. A formal comparison of the nested models was conducted using the roc.test function from the pROC R library (version 1.19.0.1)[25].

Two-sided statistical tests were considered for all analyses.

### Reporting summary

Further information on research design is available in the Nature Portfolio Reporting Summary linked to this article.

## Data availability

The source data that were analysed during the current study are not publicly available due to them containing information that could compromise participant consent and confidentiality. Data can be obtained from the data owners. Instructions for accessing Generation Scotland data can be found here: (https://www.ed.ac.uk/generation-scotland/for-researchers/access); the 'GS Access Request Form' can be downloaded from this site. Completed request forms must be sent to access@generationscotland.org to be approved by the Generation Scotland Access Committee. Source data are provided with this paper.

## Code availability

Fully annotated copies of the R analysis scripts are available at: (https://doi.org/10.5281/zenodo.17455774[26]).

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

## Acknowledgements

This research was funded in whole, or in part, by the Wellcome Trust (104036/Z/14/Z and 221890/Z/20/Z). For the purpose of open access, the author has applied a CC BY public copyright license to any Author Accepted Manuscript version arising from this submission. Generation Scotland received core support from the Chief Scientist Office of the Scottish Government Health Directorates (CZD/16/6) and the Scottish Funding Council (HR03006). DNA methylation profiling of the Generation Scotland samples was carried out by the Genetics Core Laboratory at the Edinburgh Clinical Research Facility, Edinburgh, Scotland, and was funded by the Medical Research Council UK and Wellcome (Wellcome Trust Strategic Award Stratifying Resilience and Depression Longitudinally (STRADL; Reference 104036/Z/14/Z). DNA methylation data for Generation Scotland was also funded by a 2018 NARSAD Young Investigator Grant from the Brain & Behaviour Research Foundation (Ref: 27404; awardee: Dr David M Howard) and by a John, Margaret, Alfred and Stewart Sim Fellowship from the Royal College of Physicians of Edinburgh (Awardee: Dr Heather C Whalley). D.W.B. is a fellow of the CIFAR CBD Network and is supported in part by R01AG073402 and R01AG087158. T.C and C.M. are funded by the Mayo Clinic Robert and Arlene Kogod Centre on Aging.

## Author contributions

C.M., D.L.M. and R.E.M. analysed the data, drafted the initial manuscript and designed the study. D.W.B., K.Y., M.M., V.N.G. and T.C. provided critical feedback on the study design and presentation of results. A.C., A.R. and D.L.M. were involved in the data generation. K.Y. generated the epigenetic clock data via the Biolearn platform. All authors read and approved the final manuscript.

## Competing interests

R.E.M. is an advisor to the Epigenetic Clock Development Foundation and Optima Partners Ltd. D.L.M. is employed by Optima Partners Ltd. D.W.B. is an inventor of DunedinPACE, which is licensed to Tru-Diagnostic. The remaining authors declare no competing interests.
