## [Transparent Peer Review file · Nature Communications]

An unbiased comparison of 14 epigenetic clocks in relation to 174 incident disease outcomes

Corresponding Author: Professor Riccardo Marioni

Version 0:

Reviewer comments:

Reviewer #1

(Remarks to the Author)

This manuscript makes a significant and timely contribution to the fields of epidemiology, medicine, and geroscience. It provides a large-scale analysis addressing the hotly debated topic of whether epigenetic clocks are predictive of clinical disease phenotypes and what their ultimate clinical utility might be particularly for early disease detection. To date, no comparison of the latest generation of clocks has been conducted on this scale. The study's primary strengths are its large sample size and the use of a deeply phenotyped cohort, which allowed the authors to evaluate the 10-year predictive ability of the most widely used epigenetic clocks across 174 different disease outcomes. The methodology is rigorously described, the statistical approaches appear sound and well-justified (e.g., the use of a conservative Bonferroni correction), and the results are thoughtfully contextualised in the discussion.

Furthermore, the provision of a Shiny app for results dissemination is commendable. The discussion of the study's limitations is also thorough and transparent.

Overall, I think this is a strong and valuable manuscript that provides a much-needed benchmark for the field. Addressing the points below would substantially strengthen the paper's conclusions and significantly enhance its overall impact.

Major Comments

1. Refinement of multivariate model adjustment

A key consideration is the potential for over-adjustment in the fully adjusted models. While the authors acknowledge this, the implication could be stated more forcefully: adjusting for smoking, which is a methylation proxy in clocks like GrimAge may partially attenuate the very signal the clock is designed to capture. This could lead to an underestimation of its true predictive power.

To address this and provide a clearer picture of the clocks' intrinsic predictive value, I strongly suggest the authors present results from a partially adjusted model e.g. in a supplementary table. This partially adjusted model should adjust only for core demographic covariates: age, sex, and race/ethnicity. I expect the results from this base model would be highly informative for many readers. It is crucial to distinguish this demographic adjustment from adjustment for socioeconomic factors (e.g., socioeconomic deprivation, education), which capture other aspects that are already dealt with in the fully adjusted model. The results for the fully adjusted model in Supplementary Table 2 are important and could form the focus but results for this partially adjusted model would be a valuable addition and should be briefly mentioned in the results section.

2. Quantitative summary of key findings

The results section, particularly lines 141-146, currently lacks specificity and could be more impactful. To enhance clarity, I recommend revising this section to explicitly enumerate the number/counts of significant disease associations identified for each of the leading clocks (e.g., GrimAgeV2, GrimAgeV1, DunedinPACE, PhenoAge, DNAmTL).

Following on from point 1, this quantitative summary should be provided for both the fully adjusted models and the proposed partially adjusted (age, sex, race/ethnicity) models. Presenting these counts side-by-side would allow the reader to interpret the results. A new main table summarising these counts would be highly effective, or alternatively, these counts could be incorporated into the existing Table 1.

3. Stratified analyses to address additional key questions

The authors have a valuable opportunity to conduct stratified analyses that would answer pressing questions in the field. I

would urge the authors to consider the following:

Analysis by biological sex: Do the clocks perform differently in men versus women? A comparative analysis, quantifying the number of significant disease associations in each sex, would be illuminating. Highlighting which diseases show sex-specific predictability would be a novel and important contribution to the literature.

Analysis by smoking status: A major question surrounding clocks is their utility beyond capturing smoking history. Stratifying the analysis by smoking status (e.g., never smokers versus current/former smokers) would directly address this. How many disease associations remain significant in a population of never smokers? This would provide crucial evidence for the clocks' ability to capture broader biological ageing processes independent of the strong confounding signal of smoking.

(Remarks on code availability)

Reviewer #2

(Remarks to the Author)

This paper reports on an important and comprehensive effort to evaluate the predictive utility of 14 epigenetic clocks in relation to a wide variety of disease outcomes in a very large cohort of Scottish research participants.

Because the universe of possibilities with this dataset is vast, many choices made by the authors could be questioned on various grounds. I appreciate the dilemma this presents.

Nevertheless, I wonder if some additional rationale might be offered regarding the specific choices of clocks (see Table 1). The clocks in Table 1 are described as "leading clocks" but it is somewhat unclear what is meant by "leading". Does this refer, e.g., to the number of citations to the clock in the literature? In short, could a more quantitative, evidentiary rationale be offered for this selection of clocks (e.g. citation count)?

Similarly, the choice of covariates could also benefit from additional description of the underlying rationale. For example, why is alcohol use restricted to the last week (vs. some index of lifetime exposure)? This stands in contrast to smoking, which is indexed as the more traditional "pack years" variable.

Given the size of the sample, some obvious potential moderating factors seem worth examining. For example, disease incidence differs by sex, and sex is an obvious candidate moderator variable in this analysis.

Finally, the disease outcomes are treated individually, without regard to their potentially multivariate structure. It would not be too heavy of a lift to examine outcomes together. For example, a simple count representing "multi-morbidity" could be computed and examined as an outcome.

(Remarks on code availability)

N/A

Reviewer #3

(Remarks to the Author)

The paper "An unbiased comparison of 14 epigenetic clocks in relation to 10-year onset of 174 disease outcomes in 18,859 individuals" presents to the best of my knowledge the most comprehensive comparison of existing epigenetic clocks in what is one of the largest Biobanks with available DNAm data (~18000-19000 samples). The statistical analysis is performed to a high rigorous standard. Whilst the results are in line with expectations and several previous papers comparing a subset of these clocks on smaller datasets, it is important to see that these results are validated in one of the largest population cohort DNAm datasets. So, overall, it is an important study. I only have a few suggestions for adding more analyses, in the hope that these can further strengthen this MS:

Comments:

1) Fig2: Would it be possible in Fig.2 and associated analyses, to assess if the improvement in the AUC between the full and null model is statistically significant. Same comments would also apply to the Cox-regressions. I note that the improvement in the AUC between the full and null models is generally speaking modest (~1%). Given the sample sizes involved, I presume that this is statistically significant and maybe the authors refrained from adding error bars to Fig.2, because these are too small anyway. However, this needs to be checked, as after reading Methods it is unclear if this was ever done.

2) Cell-type estimation in Biolearn: I think that there should be more clarity on this, in particular my understanding is that Biolearn is using python-adapted versions of packages or reference panels from other authors, which I feel should be duly cited. It also needs to be clarified for how many cell-types fractions were estimated? I presume 12? Which panel was used? Salas LA et al Nat Comm 2022 or Luo Q et al Genome Med 2023? Judging from the website it would seem to be Salas LA? Whilst the Salas and Luo panels yield very similar estimates (they were derived from the same sorted data), I would still recommend performing the analyses using the two different 12 cell-type panels, in order to check robustness of the results. Indeed, whilst the Salas LA et al panel includes both hypomethylated and hypermethylated markers, Luo Q et al's panel only includes cell-type hypomethylated markers, which is more in line with findings from WGBS atlases (Loyfer et al Nature) demonstrating that the most cell-type specific regions are unmethylated in the given cell-type (We have some evidence that a proportion of the hypermethylated markers in the Salas et al reference are false positives because they have very small

effect size).

3) Fig 1: If I understood correctly, the violin represents distributions over the 174 diseases. As such, we need to be careful with the error-bars shown, because if we want to rigorously compare between clocks, the correct statistical test would be a paired one (say paired Wilcoxon test), i.e we need to compare the respective values of two clocks for the same disease and across all 174 diseases. Implication is that although the difference between GrimAge2 and DunedinPACE might not seem statistically significant based on the violins shown, it could theoretically be statistically significant if say all the values for one clock are marginally higher than for the other. In this regard, I would recommend adding to Fig.1 a heatmap of one-tailed paired-test P-values between each pair of clocks, to help the reader assess the inter-clock comparisons. In case some of these turn out to be significant, it might not be a bad idea to perhaps display some of these pairwise comparisons by explicitly adding lines between datapoints corresponding to the same disease.

4) Add Episcores? I wonder if it would not be interesting to add the Episcore DNAm-proxies (which this same lab has pioneered) to these analyses, i.e. do the age-acceleration measures correlate with Episcores, and if so which ones and are these also more relevant to the respiratory and liver diseases that these 2nd/3rd generation clocks are correlating best with?

(Remarks on code availability)

R shiny application is working well.

Version 1:

Reviewer comments:

Reviewer #1

(Remarks to the Author)

Thank you for the thorough revisions. All of my comments have been fully addressed.

(Remarks on code availability)

Reviewer #2

(Remarks to the Author)

The authors were responsive to the feedback on their initial submission, and I have no further suggestions.

(Remarks on code availability)

Reviewer #3

(Remarks to the Author)

I am happy with the revisions made. Concerning the Episcores, that these were derived from the same cohort does not, in my sincere opinion, constitute a valid reason for not including them in what is a brief manuscript. However, I also don't consider this as a necessary addition.

One serious issue I spotted is with the references, as some of the citations seem to have mistakes in them e.g. ref [18] should be:

Luo Q, Dwaraka VB, Chen Q, Tong H, Zhu T, Seale K, Raffaele JM, Zheng SC, Mendez TL, Chen Y, et al: A meta-analysis of immune-cell fractions at high resolution reveals novel associations with common phenotypes and health outcomes. *Genome Med* 2023, 15:59.

and NOT (author list is wrong and muddled up)

18. Liu, Q., Dang, V.B., Chen, Q., Huang, T., Sallis, K., Rodriguez, J.M., Chen, Y., Mcrae, K., Bell, S., Lleshi, J., Magnuson, T.L., Ventham, S., Erridge, N., Zhang, T., Cardenas, N., Zheng, S.C., Saffari, R., Sliker, J.L. and Teschendorff, A.E.

(2023) 'A meta-analysis of immune-cell fractions at high resolution reveals novel associations with common phenotypes and health outcomes', *Genome Medicine*, 15, pp. 60. doi: 10.1186/s13073-023-01211-5.

Don't the authors use a reference manager like EndNote?

(Remarks on code availability)

Reviewer #1 (Remarks to the Author)

This manuscript makes a significant and timely contribution to the fields of epidemiology, medicine, and geroscience. It provides a large-scale analysis addressing the hotly debated topic of whether epigenetic clocks are predictive of clinical disease phenotypes and what their ultimate clinical utility might be particularly for early disease detection. To date, no comparison of the latest generation of clocks has been conducted on this scale. The study's primary strengths are its large sample size and the use of a deeply phenotyped cohort, which allowed the authors to evaluate the 10-year predictive ability of the most widely used epigenetic clocks across 174 different disease outcomes. The methodology is rigorously described, the statistical approaches appear sound and well-justified (e.g., the use of a conservative Bonferroni correction), and the results are thoughtfully contextualised in the discussion.

Furthermore, the provision of a Shiny app for results dissemination is commendable. The discussion of the study's limitations is also thorough and transparent.

Overall, I think this is a strong and valuable manuscript that provides a much-needed benchmark for the field. Addressing the points below would substantially strengthen the paper's conclusions and significantly enhance its overall impact.

Response: Thank you very much for the positive feedback on our manuscript, in addition to your helpful comments, which have improved the manuscript. We have responded to each point below and detail the line numbers where edits are made.

Major Comments

1. Refinement of multivariate model adjustment

A key consideration is the potential for over-adjustment in the fully adjusted models. While the authors acknowledge this, the implication could be stated more forcefully: adjusting for smoking, which is a methylation proxy in clocks like GrimAge may partially attenuate the very signal the clock is designed to capture. This could lead to an underestimation of its true predictive power.

To address this and provide a clearer picture of the clocks' intrinsic predictive value, I strongly suggest the authors present results from a partially adjusted model e.g. in a supplementary table. This partially adjusted model should adjust only for core demographic covariates: age, sex, and race/ethnicity. I expect the results from this base model would be highly informative for many readers. It is crucial to distinguish this demographic adjustment from adjustment for socioeconomic factors (e.g., socioeconomic deprivation, education), which capture other aspects that are already dealt with in the fully adjusted model. The results for the fully adjusted model in Supplementary Table 2 are important and could form the focus but results for this partially adjusted model would be a valuable addition and should be briefly mentioned in the results section.

Response: We agree with the reviewer. Initially, we presented a single model for ease of presentation. However, we now report the results from age and sex adjusted models (all participants are white European so there is no race/ethnicity variable to consider). These results have been added to Table S2.

We briefly report the new results in several places, whilst retaining our primary focus on the output from the fully adjusted analyses (new text highlighted in yellow):

Line 84: "Cox proportional hazards regression was run for each clock-disease pairing, first adjusting for just age and sex (Table S2) followed by a fully-adjusted model that also controlled for ~~adjusting for age, sex,~~ body mass index, smoking, alcohol consumption, education, and socioeconomic deprivation (Table S3). Interaction models were considered for both the basic and fully-adjusted models to test for differences by smoking status (ever/never smoked) or sex (Table S2)."

Line 102: "There were between 12 (Lin clock) and 72 (both GrimAgeV2 and DunedinPACE) statistically significant associations ($P < 0.05/174$) in the age- and sex-adjusted Cox regression models (Table 1 and Table S2)."

Line 309: "Additional analyses included basic-adjusted Cox models (controlling only for age and sex) as well as sex- and smoking (ever/never)-stratification and interaction testing for both the primary analysis (fully-adjusted Cox models) and the basic adjusted Cox models. When stratifying by smoking or testing the interaction formally, the continuous pack years variable was excluded as a covariate."

Line 314: "The proportional hazards assumption was tested by examining the Schoenfeld residuals. A test of the residuals for both the overall model and the predictor variable of interest (age acceleration) was performed using the `cox.zph` function in the survival package. This was done for both the basic- and fully-adjusted Cox models as well as for the stratified analyses (male, female, ever smoker, never smoker)."

2. Quantitative summary of key findings

The results section, particularly lines 141-146, currently lacks specificity and could be more impactful. To enhance clarity, I recommend revising this section to explicitly enumerate the number/counts of significant disease associations identified for each of the leading clocks (e.g., GrimAgeV2, GrimAgeV1, DunedinPACE, PhenoAge, DNAmTL).

Following on from point 1, this quantitative summary should be provided for both the fully adjusted models and the proposed partially adjusted (age, sex, race/ethnicity) models. Presenting these counts side-by-side would allow the reader to interpret the

results. A new main table summarising these counts would be highly effective, or alternatively, these counts could be incorporated into the existing Table 1.

Response: Thank you for highlighting this. As suggested, we have updated Table 1 to incorporate the number of significant associations (at both $P < 0.05$ and $P < 0.05/174$) for each clock in the basic- and fully-adjusted models. The full set of results are presented in Table S2.

3. Stratified analyses to address additional key questions

The authors have a valuable opportunity to conduct stratified analyses that would answer pressing questions in the field. I would urge the authors to consider the following:

Analysis by biological sex: Do the clocks perform differently in men versus women? A comparative analysis, quantifying the number of significant disease associations in each sex, would be illuminating. Highlighting which diseases show sex-specific predictability would be a novel and important contribution to the literature.

Analysis by smoking status: A major question surrounding clocks is their utility beyond capturing smoking history. Stratifying the analysis by smoking status (e.g., never smokers versus current/former smokers) would directly address this. How many disease associations remain significant in a population of never smokers? This would provide crucial evidence for the clocks' ability to capture broader biological ageing processes independent of the strong confounding signal of smoking.

Response: We agree that these are interesting analyses. We have updated the manuscript to include results stratified by sex and smoking (never vs ever) in addition to a formal comparison by interaction models. This is done for both the basic- and fully-adjusted models.

At a Bonferroni significant threshold ($P < 0.05/174$) there was minimal evidence of interactions. Of the 11 significant interactions, only 3 were from the fully adjusted models. Most interesting was the increased risk of chronic fatigue syndrome for smokers with elevated GrimAge (both v1 and v2). The other association, Ying AdaptAge by sex in relation to ptosis had small numbers ($N < 30$) of incident cases in both males and females.

Line 139: "In the fully adjusted sex- and smoking-stratified analyses and interaction models, there was two significant findings where the number of cases per stratum level exceeded 30. The association between GrimAge (both V1 and V2) and the risk of postviral fatigue syndrome was significantly higher for ever smokers compared to never smokers (HR per SD of GrimAgeV1 in ever smokers = 1.51 compared to 0.81 in never smokers, $P_{\text{interaction}} = 6.2 \times 10^{-5}$)."

Line 157: "Our findings clearly indicate that second- and third-generation epigenetic clocks should be prioritised for disease association studies. These clocks showed particularly strong links to respiratory and liver-related disease outcomes, including primary lung and oesophageal cancers and cirrhosis. Furthermore, the findings are present after adjusting for key risk factors such as deprivation, self-reported smoking behaviour and alcohol consumption as covariates. There was minimal evidence for differences in the clock-disease associations by sex or smoking status."

Reviewer #2 (Remarks to the Author):

This paper reports on an important and comprehensive effort to evaluate the predictive utility of 14 epigenetic clocks in relation to a wide variety of disease outcomes in a very large cohort of Scottish research participants.

Because the universe of possibilities with this dataset is vast, many choices made by the authors could be questioned on various grounds. I appreciate the dilemma this presents.

Response: Thank you! We found it challenging to select a parsimonious set of models to present. This point was highlighted by all three reviewers. We have thus updated the analyses to include basic-adjusted and sex/smoking stratified analyses to complement the findings from the fully adjusted regression analyses. We still focus on these latter models for the primary discussion/interpretation.

Nevertheless, I wonder if some additional rationale might be offered regarding the specific choices of clocks (see Table 1). The clocks in Table 1 are described as “leading clocks” but it is somewhat unclear what is meant by “leading”. Does this refer, e.g., to the number of citations to the clock in the literature? In short, could a more quantitative, evidentiary rationale be offered for this selection of clocks (e.g. citation count)?

Response: This is an excellent point. We have removed leading from the text. The rationale for clock selection was simply to choose those presented on the Biolearn platform. This does include the most highly cited first/second/third generation clocks (Hannum, Horvath, GrimAge, Dunedin and PhenoAge) but is not completely comprehensive. We now note this limitation in the discussion section of the paper.

Line 77: “Here, we provide an unbiased assessment of 14 ~~leading~~ clocks, including the most widely studied first, second and third generation clocks (Hannum, Horvath, PhenoAge, GrimAge and DunedinPACE, respectively; Table 1)...”

Line 171: “Our study contains certain limitations. For example, we related DNAm profiling from whole-blood samples against multi-tissue diseases and non-blood tissue specific diseases in a Scottish-based cohort. We also restricted our analyses to 14 epigenetic clocks that are presented on the Biolearn platform. Our data analysis did not exhaustively adjust for covariates...”

Similarly, the choice of covariates could also benefit from additional description of the underlying rationale. For example, why is alcohol use restricted to the last week (vs. some index of lifetime exposure)? This stands in contrast to smoking, which is indexed as the more traditional “pack years” variable.

Response: Generation Scotland participants were asked to self-report how many units of alcohol they had consumed in the prior week and whether this was about normal, more than normal or less than normal. We chose to include the units per week as a rough estimation of alcohol intake. We have added details on the self-report questionnaire to the methods section.

Line 178: "Smoking status and alcohol consumption over the past week data were self-reported and may therefore carry biases."

Line 279: "Demographic, lifestyle and health variables were considered as covariates in the regression models with each clock and disease outcome. These included: age, sex, body mass index (BMI, calculated as weight in kg divided by squared height in metres), smoking pack years (calculated by self-reported packs per day multiplied by number of years as a smoker), alcohol intake, which was assessed as units consumed over the last week in a self-report questionnaire, years of education (an ordinal variable with 11 self-report categories: (1) 0 years, (2) 1–4 years, (3) 5–9 years, (4) 10–11 years, (5) 12–13 years, (6) 14–15 years, (7) 16–17 years, (8) 18–19 years, (9) 20–21 years, (10) 22–23 years, (11) 24 or more years), and an area-based rank of socioeconomic deprivation (Scottish Index of Multiple Deprivation, SIMD). SIMD measures relative deprivation across 6,976 data zones, calculated across seven domains: income, employment, education, health, access to services, crime and housing."

Given the size of the sample, some obvious potential moderating factors seem worth examining. For example, disease incidence differs by sex, and sex is an obvious candidate moderator variable in this analysis.

Response: This is another excellent point that was also highlighted by the other reviewers. We have now included sex and smoking stratified analyses (plus interaction models). Interestingly, we found minimal evidence for significant interactions ($P < 0.05/174$).

Line 309: "Additional analyses included basic-adjusted Cox models (controlling only for age and sex) as well as sex- and smoking (ever/never)-stratification and interaction testing for both the primary analysis (fully-adjusted Cox models) and the basic adjusted Cox models. When stratifying by smoking or testing the interaction formally, the continuous pack years variable was excluded as a covariate."

Line 139: "In the fully adjusted sex- and smoking-stratified analyses and interaction models, there were only two significant findings from analyses where the number of cases per stratum level exceeded 30: The associations between GrimAge V1 and V2 with risk of post-viral fatigue syndrome (HR per SD of

GrimAgeV1 in ever smokers = 1.51 compared to 0.81 in never smokers, $P_{\text{interaction}} = 6.2 \times 10^{-5}$).

Line 157: "Our findings clearly indicate that second- and third-generation epigenetic clocks should be prioritised for disease association studies. These clocks showed particularly strong links to respiratory and liver-related disease outcomes, including primary lung and oesophageal cancers and cirrhosis. Furthermore, the findings are present after adjusting for key risk factors such as deprivation, self-reported smoking behaviour and alcohol consumption as covariates. There was minimal evidence for differences in the clock-disease associations by sex or smoking status."

Finally, the disease outcomes are treated individually, without regard to their potentially multivariate structure. It would not be too heavy of a lift to examine outcomes together. For example, a simple count representing "multi-morbidity" could be computed and examined as an outcome.

Response: Thank you for this suggestion. We have spent a lot of time thinking about how best to model multimorbidity in Generation Scotland. This includes order of disease diagnoses, time between events and how best to account for potential age/period/cohort effects. Given the complexities, we feel that these analyses are better suited to a standalone paper. Furthermore, our intention with this paper was to focus on the potential of clocks to improve the prediction of specific disease outcomes. As multimorbidity is, by definition, an amalgam of disease data, we would again prefer to leave this to a separate set of analyses. We now mention such analyses in the discussion section of the manuscript.

Line 187: "~~The time from blood draw to disease onset varied in this study. Investigations of specific time windows e.g., in increments of 5 years, could yield different strengths of associations.~~ In our effort to summarize associations across a very large number of individual diseases, we were limited in our ability to explore heterogeneity. While we consider differences by smoking status and sex, we have not explored variation across different periods of follow-up, strata of comorbidities, or stages of disease. These represent important future directions for follow-up studies, as do investigations of clock associations with metrics of aggregate disease burden, such as indices of comorbidity."

Reviewer #3 (Remarks to the Author)

The paper "An unbiased comparison of 14 epigenetic clocks in relation to 10-year onset of 174 disease outcomes in 18,859 individuals" presents to the best of my knowledge the most comprehensive comparison of existing epigenetic clocks in what is one of the largest Biobanks with available DNAm data (~18000-19000 samples). The statistical analysis is performed to a high rigorous standard. Whilst the results are in line with expectations and several previous papers comparing a subset of these clocks on smaller datasets, it is important to see that these results are validated in one of the largest population cohort DNAm datasets. So, overall, it is an important study. I only have a few suggestions for adding more analyses, in the hope that these can further strengthen this

Response: Thank you for your positive comments and for your helpful suggestions below. We have run these additional analyses, which have strengthened the manuscript.

MS:

Comments:

1) Fig2: Would it be possible in Fig.2 and associated analyses, to assess if the improvement in the AUC between the full and null model is statistically significant. Same comments would also apply to the Cox-regressions. I note that the improvement in the AUC between the full and null models is generally speaking modest (~1%). Given the sample sizes involved, I presume that this is statistically significant and maybe the authors refrained from adding error bars to Fig.2, because these are too small anyway. However, this needs to be checked, as after reading Methods it is unclear if this was ever done.

Response: Thanks for highlighting this. We have added in p-values to compare the AUC between the full and null models (added to Table S3).

Line 326: "In addition to the Cox models, logistic regression was used to generate area under the curve (AUC) estimates for 10-year disease classification. Null models contained the aforementioned covariates, while a full model also included an age acceleration measure. A formal comparison of the nested models was conducted using the roc.test function from the pROC R library."

Given these results, we have now updated Figure 2 to display the AUC differences for clock-disease associations that are

- significant in the Cox model ($P < 0.05/174$)
- nominally significant in the formal AUC comparison ($P < 0.05$)
- at least 1% increase in the AUC between the null and full model

We note this in the updated legend.

Figure 2 Legend: "**Figure 2.** AUC increments for disease – clock associations where the hazard ratio from the Cox regression was Bonferroni significant ($P < 0.05/174$), and the AUC of the full model was > 0.8 and the improvement to the AUC upon adding the clock to the null model was > 0.01 and nominally significant ($P < 0.05$) in a comparison of the nested models. Null model covariates include: age, sex, education, alcohol, smoking, BMI and deprivation. The full models also included the relevant epigenetic clock."

As Figure 2 is already quite busy with point estimates, we would prefer to illustrate the differences between the null and full models without adding confidence intervals.

Line 41: "Furthermore, there were for 35 32 of the 176 findings, instances where adding the a clock to a null classification model with traditional risk factors significantly increased the classification accuracy by $> 1\%$ with an AUCfull > 0.80 ."

Line 146: "After filtering to the 176 clock-disease associations that were Bonferroni significant in the Cox models, there were 35 32 instances where the AUC from the full model exceeded 0.8 (indicative of being clinically meaningful) and the AUC improvement between the null and full model was greater than 0.01 and nominally significant at $P < 0.05$ (Figure 2 and Table S6S8)."

2) Cell-type estimation in Biolearn: I think that there should be more clarity on this, in particular my understanding is that Biolearn is using python-adapted versions of packages or reference panels from other authors, which I feel should be duly cited. It also needs to be clarified for how many cell-types fractions were estimated? I presume 12? Which panel was used? Salas LA et al Nat Comm 2022 or Luo Q et al Genome Med 2023? Judging from the website it would seem to be Salas LA? Whilst the Salas and Luo panels yield very similar estimates (they were derived from the same sorted data), I would still recommend performing the analyses using the two different 12 cell-type panels, in order to check robustness of the results. Indeed, whilst the Salas LA et al panel includes both hypomethylated and hypermethylated markers, Luo Q et al's panel only includes cell-type hypomethylated markers, which is more in line with findings from WGBS atlases (Loyfer et al Nature) demonstrating that the most cell-type specific regions are unmethylated in the given cell-type (We have some evidence that a proportion of the hypermethylated markers in the Salas et al reference are false positives because they have very small effect size).

Response: This is another important point – thank you for raising this. We now cite the Salas paper when we introduce the WBC adjustments. We have also now

compared the age acceleration residuals when adjusting for the Salas WBCs and the EpiDISH WBCs (correlation range of 0.90-0.99).

Line 243: "Age acceleration for each clock/predictor was calculated as the residual from a linear mixed model where the clock/predictor was regressed on age and estimated cell proportions (derived from the Biolearn platform using the Salas et al. approach [16])."

Line 249: "As there are multiple approaches to estimating white cell proportions from DNA methylation data, a sensitivity analysis was conducted where each clock was adjusted for proportions derived from the EpiDISH R package [18] (reference panel set to "cent12CT.m", method="RPC"; basophil proportion was excluded to avoid collinearity as the sum of the proportions is one). The correlation between these residuals and those used in the analyses ranged between 0.90 (Zhang clock) to 0.99 (YingDamAge), suggesting minor differences between the two approaches."

3) Fig1: If I understood correctly, the violin represents distributions over the 174 diseases. As such, we need to be careful with the error-bars shown, because if we want to rigorously compare between clocks, the correct statistical test would be a paired one (say paired Wilcoxon test), i.e we need to compare the respective values of two clocks for the same disease and across all 174 diseases. Implication is that although the difference between GrimAge2 and DunedinPACE might not seem statistically significant based on the violins shown, it could theoretically be statistically significant if say all the values for one clock are marginally higher than for the other. In this regard, I would recommend adding to Fig.1 a heatmap of one-tailed paired-test P-values between each pair of clocks, to help the reader assess the inter-clock comparisons. In case some of these turn out to be significant, it might not be a bad idea to perhaps display some of these pairwise comparisons by explicitly adding lines between datapoints corresponding to the same disease.

Response: Thank you for this suggestion. For these analyses we compared effect sizes across all 174 diseases (with disease label as a random effect) with GrimAge v1 as the reference category. We have now run additional analyses (paired t-tests) to compare each pairwise combination of clocks and present the difference in mean effect size and its statistical significance. We have added a new panel to Figure 1 to display these findings. They illustrate similar findings whereby the second/third generation clocks outperform the first generation measures.

Line 319: "A formal comparison of the log Hazard Ratios for the clocks across all 174 disease outcomes in the fully adjusted model (primary analyses) was then

conducted using a linear mixed effects model [20,21]. The logHR was the outcome while the clock was the predictor (GrimAge v1 was the reference level) and disease was included as a random effect on the intercept. The emm function from the emmeans R package [22] was then used to obtain the marginal means for each clock, prior to formal pairwise comparisons with a Tukey adjustment to account for multiple comparisons."

Figure 1. (A) Distribution of log hazard ratios for each epigenetic clock across all 174 incident disease outcomes. Clocks are presented in descending order of the median effect size. Median and interquartile range presented within the violins. First generation clocks are highlighted in pink, second generation in green, third generation clocks in purple and telomere length (where log HR effect sizes have been multiplied by -1 for visual display purposes) in turquoise. **(B)** Comparison of average log hazard ratios across 174 disease outcomes across pairs of epigenetic clocks. Telomere length log hazards have been multiplied by -1 prior to comparisons. The mean differences are presented as x-axis clock minus y-axis clock e.g., there is a negative difference between DNAmTL and DunedinPACE, indicating the mean log hazards for the latter are larger. * P<0.05, ** P<0.01, *** P<0.001

4) Add Episcopes? I wonder if it would not be interesting to add the Episcopes DNAm-proxies (which this same lab has pioneered) to these analyses, i.e. do the age-acceleration measures correlate with Episcopes, and if so which ones and are these also more relevant to the respiratory and liver diseases that these 2nd/3rd generation clocks are correlating best with?

Response: This is an excellent idea and something we are very keen to explore. However, the majority of EpiScores that we have created have been trained in the same Generation Scotland population, which would lead to overfitting. Given that the focus of this paper is the link between epigenetic clocks and disease, we also feel that the inclusion of EpiScores (e.g., those from our *elife* publication: PMID 35023833) would distract from the main message, especially given that we would need to run basic and fully adjusted models, with both sex and smoking interactions, generating tens of thousands of new results.